

# FastViromeExplorer: a pipeline for virus and phage identification and abundance profiling in metagenomics data

Saima Sultana Tithi[1], Frank O. Aylward[2], Roderick V. Jensen[2] and Liqing Zhang[1]

[1] Department of Computer Science, Virginia Polytechnic Institute and State University (Virginia Tech), Blacksburg, VA, United States of America
[2] Department of Biological Sciences, Virginia Polytechnic Institute and State University (Virginia Tech), Blacksburg, VA, United States of America

## ABSTRACT

With the increase in the availability of metagenomic data generated by next generation sequencing, there is an urgent need for fast and accurate tools for identifying viruses in host-associated and environmental samples. In this paper, we developed a stand-alone pipeline called FastViromeExplorer for the detection and abundance quantification of viruses and phages in large metagenomic datasets by performing rapid searches of virus and phage sequence databases. Both simulated and real data from human microbiome and ocean environmental samples are used to validate FastViromeExplorer as a reliable tool to quickly and accurately identify viruses and their abundances in large datasets.

## INTRODUCTION

Identifying the kinds of viruses that infect eukaryotes and prokaryotes (phages) and understanding their functions are important because they are the most abundant entities on Earth (*Paez-Espino et al., 2016*). In the human body there are estimated to be 100 times more viral particles than eukaryotic cells (*Fancello, Raoult & Desnues, 2012*). Studies have shown that there are connections between human gut microbiome (viruses and bacteria) and diseases such as inflammatory bowel disease (IBD) and colorectal cancer (*Mills et al., 2013*; *Mirzaei & Maurice, 2017*; *Hannigan et al., 2017*). Moreover, recent emerging viral outbreaks including the Zika outbreak in Brazil (*Campos, Bandeira & Sardi, 2015*), Ebola in West Africa (*Carroll et al., 2015*; *Gire et al., 2014*), the Middle East respiratory syndrome coronavirus (MERS-CoV) (*Haagmans et al., 2014*), SARS and influenza-A caused tens of thousands of human deaths. To better understand and eventually prevent such viral outbreaks, it is critical to have timely identification and annotation of viruses. In addition, viruses have been shown to play important roles in shaping the composition and function of environmental microbiomes (*Rohwer & Thurber, 2009*). Traditional techniques of virus identification rely on isolation and culturing, which is not only time-consuming but often infeasible as many viruses and their hosts are difficult to cultivate in laboratories. Thanks to

Corresponding authors
Saima Sultana Tithi, saima5@vt.edu
Liqing Zhang, lqzhang@vt.edu

the fast development of biotechnology, it is now easy and quick to produce metagenomics data for a direct analysis of genetic materials to identify viruses and their abundances in various environments (*Handelsman et al., 1998*).

However, with the ease of metagenomics data generation also comes the challenge of downstream data analysis, including the computational identification of viral species and their abundances in a fast yet accurate manner from hundreds of millions/billions of short sequences. Strategies to identify and annotate viruses vary among different tools, ranging from analyzing marker genes, binning sequences or reads into taxonomic groups, assembling sequences into contigs and then annotating the genes from the contigs for taxonomy, to directly aligning short reads to a reference database and inferring virus types and abundances based on the alignment results. The most straightforward and fastest approach for virus taxonomic annotation is to align short reads to a marker gene database and identify viruses based on the alignments; for example, MetaPhlAn (*Segata et al., 2012*) and its updated version MetaPhlAn2 (*Truong et al., 2015*) use this approach. However, the marker gene analysis strategy does not work well when the input data contain species that do not have known marker genes. Comparatively, assembling reads into longer contigs and then performing the taxonomic analysis with contigs tend to produce more accurate results (*Roux et al., 2017*). This type of virus analysis pipelines normally requires the users to assemble the reads using an independent assembler and then annotates the assembled contigs (e.g., VirSorter (*Roux et al., 2015*), VirFinder (*Ren et al., 2017*), Metavir (*Roux et al., 2011*), Metavir2 (*Roux et al., 2014*), and Virome (*Wommack et al., 2012*)). Another assembly-based workflow for identifying viral elements from metagenomic reads is FRAP (fragment recruitment, assembly, purification) (*Cobián Güemes et al., 2016*). Understandably the assembly of short reads into contigs gives longer sequences including longer coding regions with more informative content, which leads to improved annotation and downstream analysis. However, read assembly can be very time-consuming for large metagenomics data and can also generate chimeras (i.e., sequences from different genomes that are incorrectly assembled together due to their similarity) that mislead downstream annotation (*Vázquez-Castellanos et al., 2014*; *Van der Walt et al., 2017*). Finally, tools such as MG-RAST (*Meyer et al., 2008*), ViromeScan (*Rampelli et al., 2016*), VIP (*Li et al., 2016*), and HoloVir (*Laffy et al., 2016*) directly align short reads to a reference database of whole genomes for taxonomy annotation. Many of these tools were initially developed for bacteria but adapted later for viruses and tend to work poorly due to the much smaller reference databases available for viruses than for bacteria (*Fancello, Raoult & Desnues, 2012*). In addition, as many virus annotation tools (i.e., Metavir (*Roux et al., 2011*), Metavir2 (*Roux et al., 2014*), Virome (*Wommack et al., 2012*), MG-RAST (*Meyer et al., 2008*)) are web-based, users need to upload their data to the website and wait for a long time to get results.

To provide fast and accurate virus detection and quantification on metagenomics data, we developed a stand-alone pipeline, FastViromeExplorer. Instead of the traditional read alignment tools such as BLAST (*Altschul et al., 1997*) or Bowtie2 (*Langmead & Salzberg, 2012*), FastViromeExplorer uses kallisto (*Bray et al., 2016*), a pseudoalignment based approach originally developed for alignment and quantification of RNA-seq data. Kallisto

has also been used to map metagenome reads to a database of bacterial genomes (*Schaeffer et al., 2017*). Here we first use kallisto to rapidly map short metagenome reads to a reference virus database. Then FastViromeExplorer filters the alignment results based on minimal coverage criteria and reports virus types and abundances along with taxonomic annotation. To test the performance of FastViromeExplorer, we used simulated datasets of a known mixture of viral, phage, and bacterial genomes with different error/mutation rates. We also applied FastViromeExplorer to real metagenome datasets generated from a Fecal Microbiota Transplantation (FMT) experiment (*Lee et al., 2017*) and from environmental ocean samples (*Aylward et al., 2017*). FastViromeExplorer is directly compared with blastn and ViromeScan (*Rampelli et al., 2016*), a recently developed read based annotation tool for eukaryotic viruses.

FastViromeExplorer is freely available at https://code.vt.edu/saima5/FastViromeExplorer.

## METHODS

FastViromeExplorer, written in Java, has two main steps: (1) the read mapping step where all reads are mapped to a reference database, and (2) the filtering step where the mapping results are subjected to three major filters (detailed later) for output of the final results on virus types and abundances. The input of the read alignment step is raw reads (single-end or paired-end) in fastq format. FastViromeExplorer uses the reference database downloaded from NCBI containing 8,957 RefSeq viral genomes as default but can also use any updated or customized databases as reference. FastViromeExplorer incorporates the reference database as an input parameter, so that user can use any database of his choice as input. A precomputed kallisto index file, generated for the 8,957 genomes is distributed here: http://bench.cs.vt.edu/FastViromeExplorer/.

First, FastViromeExplorer calls kallisto (*Bray et al., 2016*) as a subprocess to map the input reads against the reference database. Kallisto was developed to map RNA-seq data to a reference transcriptome (all the transcripts for a genome) leveraging the pseudoalignment process and estimate the abundance of the transcripts using the Expectation-Maximization (EM) algorithm (*Dempster, Laird & Rubin, 1977*). As there is no actual sequence alignment of the entire read over the reference sequences, the pseudoalignment process enables read mapping to be both lightweight and superfast. Essentially, kallisto searches for exact matches for a short $k$-mer (default size 31 bp) between the metagenomic reads and the sequences in the virus/phage database. For example, kallisto was able to map and quantify 30 million paired-end RNA-seq reads from a human transcriptome sample in less than 10 min on a small laptop computer with a 1.3-GHz processor (*Bray et al., 2016*). In addition to the ultrafast speed, kallisto also gives accurate estimation of abundance of each transcript or reference sequence (*Schaeffer et al., 2017*; *Soneson et al., 2016*). Consequently, kallisto could provide an ideal tool for detection and quantification of viruses in metagenomic samples that commonly have tens of millions of reads, mapping of which using commonly used programs such as BLAST can be time-consuming and often infeasible without computer clusters. Therefore, FastViromeExplorer deploys kallisto for the purpose of read mapping and abundance estimation of the viruses. Since kallisto searches for exact matches for a
short $k$-mer (default size 31 bp) between the metagenomic reads and the sequences in the virus/phage database, if a 31 bp match is found then the virus is detected. If multiple hits occur, then kallisto uses an EM algorithm to help resolve the redundancy and quantify the abundances of the detected viruses. The $k$-mer size in kallisto can be altered depending on user's need. For example, if the sample is expected to contain viral sequences that are divergent from those in the reference database the $k$-mer size can be reduced to improve detection sensitivity.

After the first alignment step, FastViromeExplorer takes the output of kallisto that includes information of the aligned reads together with estimated abundances or estimated read counts of all the identified viruses for the processing of the second step. The second step filters the output of the first step using three criteria, introduced to ensure the quality of virus detection and especially to reduce the number of false positive viruses from the result. In detail, the first criterion, hereafter referred to as "$R$", is based on the ratio of the observed extent of genome coverage with the expected extent of genome coverage, computed as

$$R = \frac{C_o}{C_e}, \tag{1}$$

$C_o$ is the observed extent of genome coverage by the mapped reads, computed as

$$C_o = \frac{L_s}{L_g}, \tag{2}$$

where $L_s$ is the actual length of the genome that is supported or covered by the mapped reads and $L_g$ is the length of the genome. $C_e$ is the expected extent of genome coverage, assuming a Poisson distribution for the mapped reads along the genome, and therefore,

$$C_e = 1 - e^{-\frac{N*L_r}{L_g}}, \tag{3}$$

where $N$ is the number of mapped reads to the genome, $L_r$ is the read length, and $L_g$ is the length of the genome. If a virus has $R < 0.3$, FastViromeExplorer discards the virus. This criterion is motivated by the observation that some viruses detected by our tool only have reads mapped to the repeat regions of their genomes. For example, while analyzing the fecal samples from *Lee et al. (2017)*, we found that for the BeAn 58058 virus (NC_032111.1), all the reads were mapped to one particular region of its genome, from 8,200 bp to 8,700 bp (see Fig. S1). Analyzing this region using RepeatMasker (*Smit, Hubley & Green, 1996*) revealed that it is a simple repeat region and falls into the class of Alu elements. If the virus is truly present in the sample, we expect reads to be mapped to not only the repeat region but also other regions of the genome. Therefore, finding this virus is likely an artifact caused by the prevalence of repeat regions instead of real biological signals. If the reads are all mapped to a repeat region, the observed coverage of the virus genome $C_o$ is expected to be much lower than $C_e$, as a result, $R$ is low and by imposing a cutoff of 0.3 (determined based on our empirical analyses), viruses that have reads mapped to only repeat regions get filtered out.

The second criterion requires $C_o \geq 0.1$; that is, a virus that has $C_o < 0.1$ is discarded. This criterion requires that the mapped reads should cover at least 10% of the viral genome. Manual inspection of the results of our tool reveals that very large viruses may have several repeat regions in their genomes and as a result, though all the reads are mapped to the repeat regions, they are mapped to different repeat regions. In these cases, the difference between $C_o$ and $C_e$ may be small and therefore $R$ can be high enough to pass the first filter. However, it is very likely that the result is simply an artifact of repetitive sequences. For example, while analyzing the fecal samples (*Lee et al., 2017*), we found that Pandoravirus dulcis (NC_021858.1), a very large virus with 1,908,524 bp, has several repeat regions, and all the reads were mapped only to the repeat regions (see Fig. S2). Hence, to alleviate this artifact, $C_o \geq 0.1$ is used as the second filter. As repeat regions of a virus usually cover less than 10% of the genome (*Philippe et al., 2013*), if any virus is covered by more than 10% by the reads, it is reasonable to assume that the reads are not merely from repeat regions and thus the virus should be considered in the result.

The third criterion is based on the number of mapped reads $N$. Extensive empirical analysis and inspection of the results of our tool show that for very small viruses, only a few reads are enough to cover a good portion of the viral genome, resulting in high $R$ and $C_o$ that pass criteria 1 and 2. For example, in the fecal samples (*Lee et al., 2017*) that we analyzed, four reads were mapped to Rose rosette virus RNA3 (NC_015300.1). As the viral sequence has only 1,544 bp, four reads of length 150 bp were enough to pass criteria 1 and 2. But as only a handful of reads are mapped, it is likely that the virus is false positive. To be more stringent, FastViromeExplorer applies the third filter requiring the number of mapped reads to be greater than 10, and therefore discards the ones with $N < 10$.

After applying all the filters, FastViromeExplorer outputs the final result that contains a list of identified viruses in the given sample along with the estimated read count or abundance and taxonomy of the viruses. The output list is sorted by the abundance with the most abundant viruses on the top of the list.

It is worth noting that the three criteria are introduced to improve the virus detection specificity by alleviating artifacts caused by factors such as repeat sequences and low genome coverage. The actual cutoff values for $R$, $C_o$, and $N$ are based on our empirical experience and literature observation. However, depending on the specific studies and the need of users, the cutoff values used here might not be suitable. To allow flexibility and customization, FastViromeExplorer incorporates these three filters as parameters so that users can easily adjust the values to adapt to their own studies. For example, users can deploy more stringent criteria by setting higher values for $R$, $C_o$, and $N$ than the default, to get a "high confidence" set of viruses or can lower these values to increase sensitivity to detect divergent viruses or viral reads in metagenomic data where coverage may be expected to not be uniform (*Solonenko et al., 2013*).

FastViromeExplorer was run on both simulated and real data to examine its running time and accuracy. FastViromeExplorer used kallisto (version 0.43.1) with default settings and generated pseudoalignment results in sam format and filtered abundance results in a tab-delimited file. The abundance results contain identified virus names, NCBI accession numbers, NCBI taxonomic path, and estimated read counts. FastViromeExplorer was

run on two different reference databases, the default database distributed together with FastViromeExplorer, that is, the NCBI RefSeq database containing 8,957 genomes of eukaryotic viruses and phages, and the set of sequences collected from the JGI "earth virome" study (*Paez-Espino et al., 2016*) containing 125,842 metagenomic viral contigs (mVCs). The taxonomic annotation and host information for these mVCs were collected from the IMG/VR database (*Paez-Espino et al., 2017*).

In addition to the challenge of mapping 10s or 100s of millions of metagenomic reads, tools for the accurate identification and quantification of viral genomes must also be capable of handling ever-growing reference databases of viral sequences. In order to measure how the indexing step of kallisto scales with reference databases of different sizes, kallisto was applied to index five different databases. Three databases were generated from NCBI RefSeq viral database, one containing only phages (2,187 phage genomes), one containing only eukaryotic viruses (6,770 eukaryotic virus genomes), and one containing both phages and eukaryotic viruses (8,957 viral genomes). The other two databases were created from sequences collected from *Paez-Espino et al. (2016)*, one containing all the 125,842 mVCs and the other containing half of the mVCs. The time analysis of kallisto's indexing step was produced on a Linux based cluster with 64 CPUs and 128 GB RAM. The indexing step was run using default *k*-mer size 31 and default number of threads 1. The precomputed kallisto index file for the full 125,842 mVCs from JGI is available here: http://bench.cs.vt.edu/FastViromeExplorer/.

To evaluate the performance of FastViromeExplorer, we compared speed and accuracy with ViromeScan, a recently developed virus annotation pipeline that calls Bowtie2 as a subprocess for read mapping, that was shown to be 1,000 times faster than previous tools (*Rampelli et al., 2016*). ViromeScan was run with default settings and with the eukaryotic DNA/RNA virus database containing 4,370 genome sequences, the largest reference database provided by ViromeScan, and with a custom database consisting of the 125,842 mVCs from JGI. ViromeScan generated alignment results and abundances of viruses at family, genus, and species level. We also ran blastn (version ncbi-blast-2.6.0 +) using both the NCBI RefSeq viral database and the large JGI database. Blastn only generated the alignment result in text format. All the time analyses were calculated using elapsed real time from Unix's time command.

To examine the virus detection and quantification accuracy of FastViromeExplorer, simulated metagenomic data were used. A randomly selected collection of genomes containing 4,000 virus genomes and 2,000 bacteria genomes were obtained from NCBI RefSeq database. Four paired-end read datasets, each containing one million reads of length 100 bp, were generated from these genomes using the read simulator WGSIM (https://github.com/lh3/wgsim). For all the datasets, 49% reads were from viruses and 51% from bacteria. The four datasets were generated using 1% sequencing error rate and 3%, 5%, 7%, or 10% mutation frequencies respectively. ViromeScan and blastn were also applied to these four datasets. As ViromeScan uses eukaryotic viruses as the reference database, for comparison, both FastViromeExplorer and blastn were run on a reference database containing only NCBI RefSeq eukaryotic viruses. ViromeScan was run with the eukaryotic virus database provided by ViromeScan. Under the default setting,

ViromeScan removed all the mapped reads during its quality filtering and trimming step (trimBWAstyle.pl script) and did not produce any results. Therefore, it was run without ViromeScan's quality filtering and trimming step. With the ground truth for the alignment of the reads, recall, precision, and $F1$ score were calculated using the following formula:

$$\text{Recall} = \frac{TP}{TP + FN}, \tag{4}$$

$$\text{Precision} = \frac{TP}{TP + FP}, \tag{5}$$

$$F1 \text{ score} = \frac{2 * \text{Recall} * \text{Precision}}{\text{Recall} + \text{Precision}}. \tag{6}$$

To examine the running time and performance of FastViromeExplorer in detecting viruses on real data, the fecal metagenomics datasets described in *Lee et al. (2017)* were downloaded from NCBI under the accession number SRP093449 and annotated with both FastViromeExplorer and ViromeScan. The study tracked bacteria colonization in a fecal microbiota transplantation (FMT) experiment through the analysis of metagenomic data. To examine how the viruses/bacteriophages were affected by the transplantation, we reanalyzed the four fecal metagenomic samples collected from a healthy donor and three samples from a recipient patient suffering mild/moderate ulcerative colitis. The three samples for the recipient were collected prior to FMT, four weeks after FMT, and eight weeks after FMT, respectively. All the reads were Illumina paired-end reads with 150 bp read length. Seven data sets of different sizes (1, 3, 5, 10, 20, 30, and 40 million reads) were also generated from the samples and annotated by FastViromeExplorer and ViromeScan to compare their running time on large datasets. To examine the effect of the reference database on results, FastViromeExplorer was applied to the samples using two different reference databases, FastViromeExplorer's default reference database and the set of 125,842 mVCs collected from the study *Paez-Espino et al. (2016)*. While using the NCBI RefSeq database as reference, a Linux based laptop with Intel core i5-3230M CPU @ 2.60 GHz * 4 processors and 12 GB RAM was used to produce the results, and while using the 125,842 mVCs as reference, a Linux based cluster with 64 CPUs and 128 GB RAM was used to produce the results. While using the cluster, only one thread was used to run the tools.

To examine the applicability of FastViromeExplorer on environmental samples, an ocean water metagenome file described in *Aylward et al. (2017)* was downloaded from NCBI SRA under the accession number SRX2912986 and analyzed with FastViromeExplorer. The metagenome sequencing file had around 18 million paired-end reads and the 125,842 mVCs collected from the study *Paez-Espino et al. (2016)* was used as reference database. As the original study focused on ocean virome, a viral contig set collected from Global Ocean Virome (GOV) study (*Roux et al., 2016*) was also used as reference database. The GOV contig set contains 298,383 epipelagic and mesopelagic viral contigs and a precomputed kallisto index file for this viral contig set is available here: http://bench.cs.vt.edu/FastViromeExplorer/.

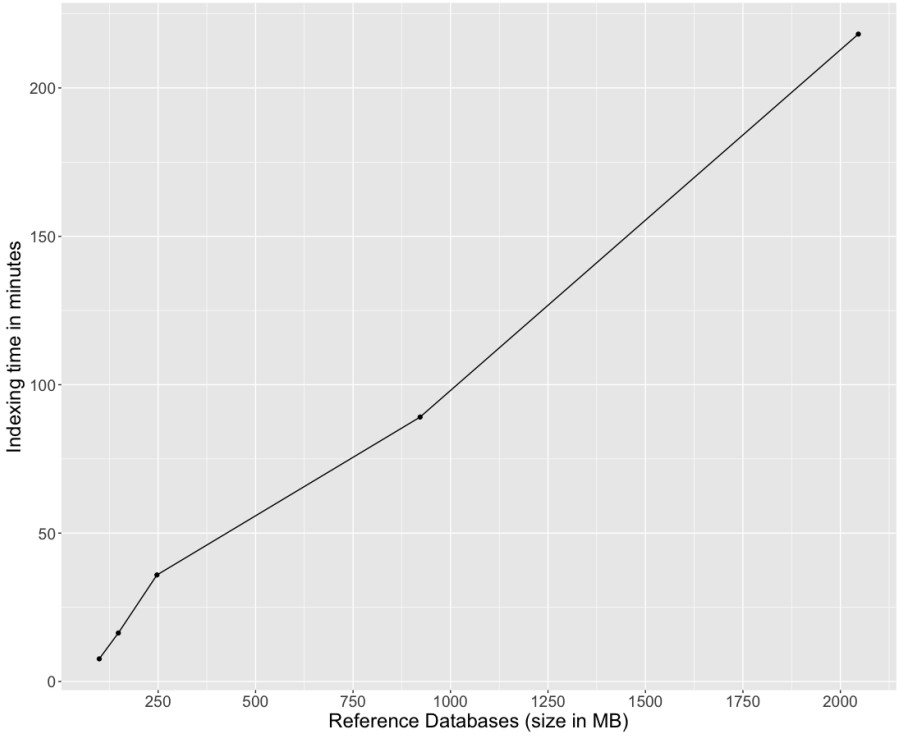

**Figure 1** Kallisto's indexing time for five reference databases, NCBI RefSeq eukaryotic viruses (99 MB), NCBI RefSeq phages (148 MB), all NCBI RefSeq viruses and phages (247 MB), 62,921 mVCs (992 MB), and 125,842 mVCs (2 GB).

## RESULTS AND DISCUSSION

We applied kallisto to index five databases of different sizes and calculated the running time of the indexing step. Figure 1 shows that indexing time increases linearly with the size of the reference databases, and for the largest reference database of 2 GB, kallisto took 3 h and 38 min to generate the index file.

To examine how running time changes with sample size, we created seven data sets with 1, 3, 5, 10, 20, 30, and 40 million reads respectively from the data described in *Lee et al. (2017)* and applied FastViromeExplorer, ViromeScan, and blastn. As blastn took too long to run on large data sets, we run blastn on only three data sets of size 1, 3, and 5 million reads respectively. Two databases, one containing all NCBI RefSeq viral genomes and the other containing 125,842 mVCs from *Paez-Espino et al. (2016)*, were used as the reference databases, to also examine the effect of reference databases on running time. Figure 2A shows the running time using the NCBI database as reference. FastViromeExplorer has the shortest running time for all the seven data sets. For the data set with 5 million reads FastViromeExplorer took only seven minutes, compared to 12 min for ViromeScan, 31 min for blastn. The speedup of FastViromeExplorer compared to ViromeScan became much more pronounced when a larger reference database was used. Figure 2B shows that when we used the larger reference database, for a data set with 5 million reads, FastViromeExplorer
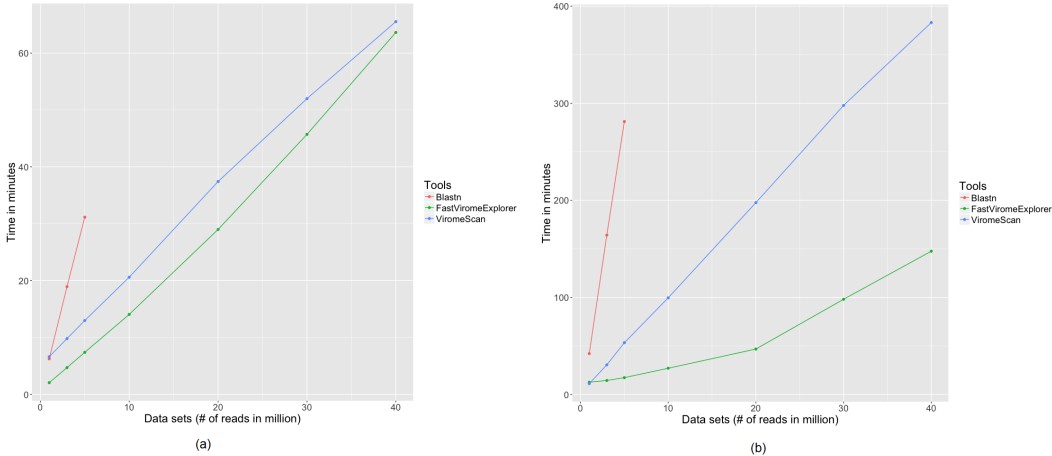

**Figure 2** Comparison of running time among FastViromeExplorer, ViromeScan, and Blastn for seven data sets with 1, 3, 5, 10, 20, 30, and 40 million reads, respectively (A) against a reference database containing 8,957 NCBI RefSeq viruses, (B) against a reference database containing 125,842 mVCs.

took 17 min, compared to 53 min for ViromeScan, and 4 h and 40 min for blastn. So FastViromeExplorer ran three times faster than ViromeScan and 16 times faster than blastn. For the largest data set with 40 million reads, FastViromeExplorer took 2 h and 27 min, a 2.5× speedup compared to ViromeScan that took 6 h and 23 min. Taken together, when using NCBI virus and phage database as reference, FastViromeExplorer takes on average about 1 min to process one million reads; when using a larger database (125,842 mVCs, 2GB), FastViromeExplorer takes 3–4 min to process one million reads, a 2–3× speed up compared to ViromeScan. Note that the indexing time (for both FastViromeExplorer and ViromeScan) was not counted in the running time shown (Fig. 2) as indexing needs to be computed only once. Once the index file is generated, it can be used to analyze any metagenomic data.

Simulated datasets were initially used to compare the annotation performance of FastViromeExplorer with ViromeScan and blastn. Since viruses mutate fast, even if it is the same viral species, the viral sequences in the metagenomic data might not be exactly the same as their sequences in the reference database, it is therefore important to examine the performance of a virus detection tool taking into account virus's high mutation rate. We therefore simulated four data sets with different mutation frequencies (3%, 5%, 7%, and 10%) from the references and applied FastViromeExplorer, ViromeScan, and blastn. Figure 3 shows the $F1$ score (*Recall* and *Precision* are given in Table S1). All the tools have had high *Precision* (99%) across all the data sets. But as mutation frequency becomes higher, the number of mapped reads is reduced and *Recall* becomes lower for all the tools. In terms of $F1$ score, blastn has the best score, FastViromeExplorer has similar but slightly lower score, and ViromeScan has the lowest score. For the data set with the highest mutation frequency (10%), the $F1$ scores for blastn, FastViromeExplorer and ViromeScan are 0.79, 0.7, and 0.43 respectively. But FastViromeExplorer took 2 min compared to blastn which took 8 min. Therefore, for these simulated data sets and using all eukaryotic viruses

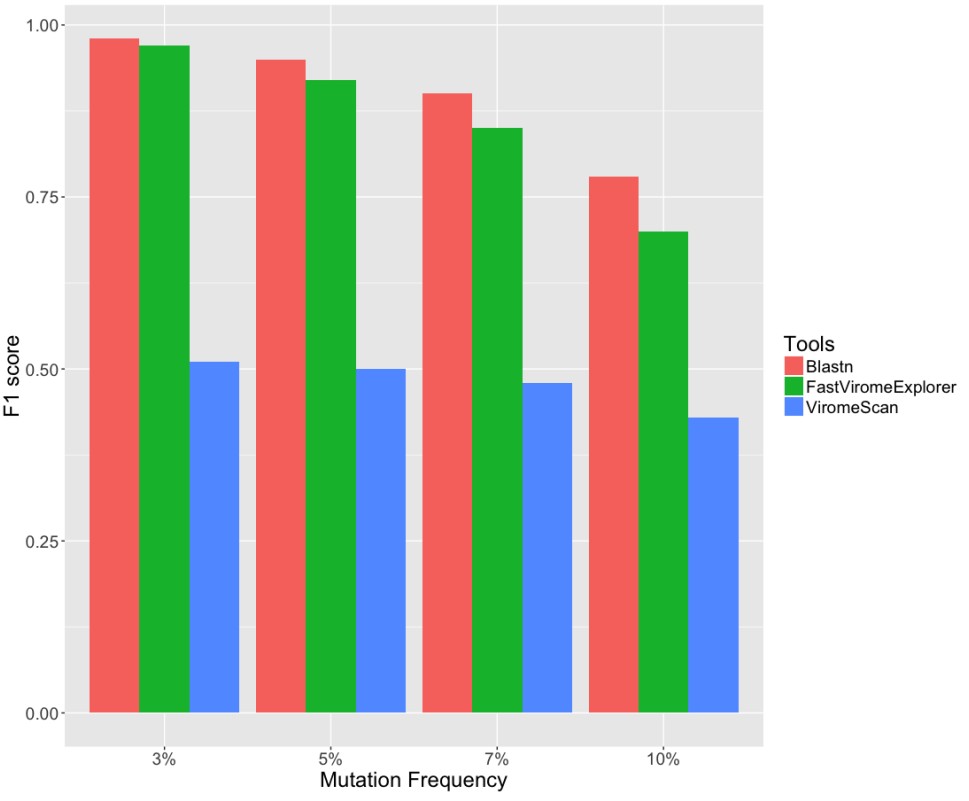

**Figure 3** *F*1 score of FastViromeExplorer, ViromeScan, and Blastn when using NCBI eukaryotic viruses as the reference database and four simulated data sets of 1 million reads each with mutation frequency 3%, 5%, 7%, and 10% respectively.

as the reference database, FastViromeExplorer runs four times faster than blastn while maintaining a similar *F*1 score to blastn.

To examine the performance of FastViromeExplorer in detecting and quantifying viruses on real data, we applied FastViromeExplorer to the fecal metagenomic samples collected from *Lee et al. (2017)*. Lee et al. followed the dynamics and consequence of fecal microbiota transplantation (FMT) by examining the metagenomics data from a donor's and recipients' preFMT and postFMT samples. They constructed 92 bacterial metagenome-assembled genomes (MAGs) from reads of the donor samples and examined the occurrence of the MAGs in the recipient samples. They found that the bacterial MAGs that were present in the donor samples and also colonized the recipient samples after FMT mostly belonged to the order *Bacteroidales*. Here we examined the dynamics of viruses/phages to see whether it is consistent with the finding of *Lee et al. (2017)*.

From the result of FastViromeExplorer using the 8,957 NCBI RefSeq viral genomes as reference, we observed that only three viruses (Human endogenous retrovirus K113, Glypta fumiferanae ichnovirus segment C10, and Lactococcus prophage bIL311) were found in all four donor samples, with human endogenous retrovirus K113 being the most abundant for samples 1, 3, and 4, and Lactococcus prophage bIL311 the most abundant in sample 2. For

the recipient, 30 viruses were found in the preFMT sample whereas only five were found in the two postFMT samples. Among the five viruses, only Lactococcus prophage was also found in one donor sample. But as this prophage was also present in the preFMT sample, we cannot conclude that the virus was transferred from the donor to the recipient. Overall, using the NCBI RefSeq database as the reference, we only detected 38 viruses in the FMT samples, and this result reveals no clear evidence of virus/phage transfer from the donor to the recipient. This result indicates that as our tool is a reference-based virus detection tool, having a suitable and/or complete reference database is important for performance.

We also applied ViromeScan to the fecal samples with its default reference database containing 4,370 eukaryotic DNA/RNA viruses. ViromeScan identified 847 viruses in all the samples. Compared to ViromeScan's reference database, ours is two times bigger and it is thus surprising that ViromeScan identified a lot more viruses than FastViromeExplorer. Analysis of the ViromeScan result shows that the most abundant virus, Encephalomyocarditis virus, has all the reads mapped to a repeat region of its genome (see Fig. S3), indicating that the annotation is likely false positive. In fact, Encephalomyocarditis virus was also present in the initial result produced by FastViromeExplorer, but was discarded after the first filtering step. To further examine the effect of our three filtering criteria, we applied them to the ViromeScan result. Figure 4 shows that most of the viruses were filtered out and only Human endogenous retrovirus K113 and Glypta fumiferanae ichnovirus remained, both of which were also present in the final result of FastViromeExplorer. The finding here shows the importance of the filtering criteria in removing viruses that might be annotation artifacts caused by repeats, low coverage, and small genome sizes.

Since the analysis of the fecal samples using the default NCBI viral database did not reveal anything meaningful about fecal microbiota transplantation from the donor to the recipient, we tried FastViromeExplorer again using the 125,842 metagenomic viral contigs (mVCs) collected from *Paez-Espino et al. (2016)* as reference. These mVCs are mostly unknown partial or complete viral genomes but have been predicted/annotated for their possible hosts and the host information of the mVCs is made available through the IMG/VR website (*Paez-Espino et al., 2016*; *Paez-Espino et al., 2017*). Therefore, the predicted host information of the mVCs, collected from the IMG/VR website, can be used to examine the result. Using these mVCs as reference, our tool detected 3,479 viral contigs in the FMT samples. Figure 5 shows the relative abundance of host bacteria across all donor and recipient samples. The order *Bacteroidales* is more abundant than the order *Clostridiales* in all donor samples. For the recipient, prior to FMT, the order *Clostridiales* clearly dominated the microbiota, however, after the transplantation, the abundance of phages infecting the order *Bacteroidales* increased dramatically and the abundance of the order *Clostridiales* decreased greatly. This result indicates that phages with host bacteria from the order *Bacteroidales* were either successfully transferred or greatly enriched as a result of the microbiota transplantation from the donor to the recipient. For example, in donor samples, "SRS049900_LANL_scaffold_14438" is one of the most abundant mVC, being the most abundant in donor samples 1 and 2, and the second most abundant in samples 3 and 4. This mVC was not present in the recipient's preFMT sample but was
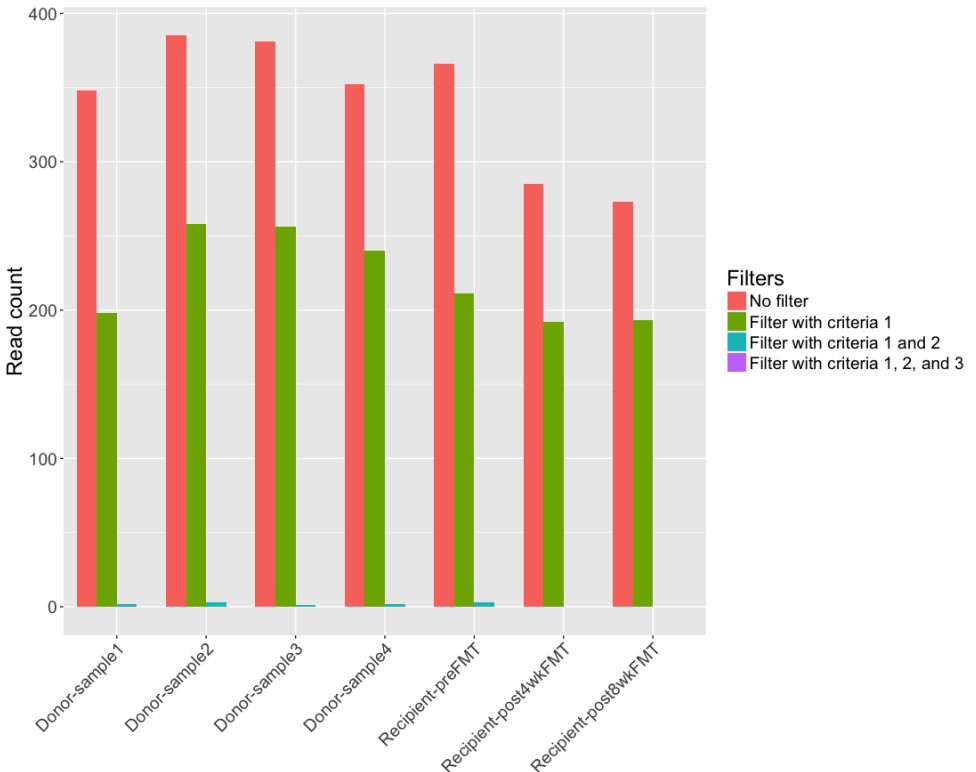

**Figure 4** Number of viruses from ViromeScan result before applying any filter, after applying criterion 1, after applying criteria 1 and 2, and after applying all three criteria.

highly abundant in the postFMT samples, suggesting either the successful transferring of the mVC from the donor to the recipient or the great enrichment of the mVCs in the recipient as a result of FMT. As the host of this mVC is from the order *Bacteroidales*, this suggests the successful colonization of bacteria from the order *Bacteroidales* from the donor to the recipient. Therefore, our result on phage transfer following the FMT is consistent with the observation on bacterial colonization following the FMT shown in the original study (*Lee et al., 2017*). The detailed annotation result is given in Table S2.

Consequently, when we applied FastViromeExplorer to the samples using a larger reference database, our tool detected 3,479 viral contigs which was much greater than the number of viruses detected using NCBI RefSeq database (38 viruses). Using a larger reference database, a much clearer correlation between our results and the biological results reported in the original paper emerges, highlighting the importance of having larger and more complete reference databases.

We also applied FastViromeExplorer to ocean microbiome samples collected at multiple time points from *Aylward et al. (2017)*. This study assembled 483 viral scaffolds (NCBI accession numbers NTLX01000001.1–NTLX01000483.1) from the metagenome reads from 44 ocean samples. In our study, we tried to find if FastViromeExplorer could rapidly identify dominant viral components directly from the read files using both the JGI 125,842 mVC data set collected from *Paez-Espino et al. (2016)* as well as the GOV dataset containing

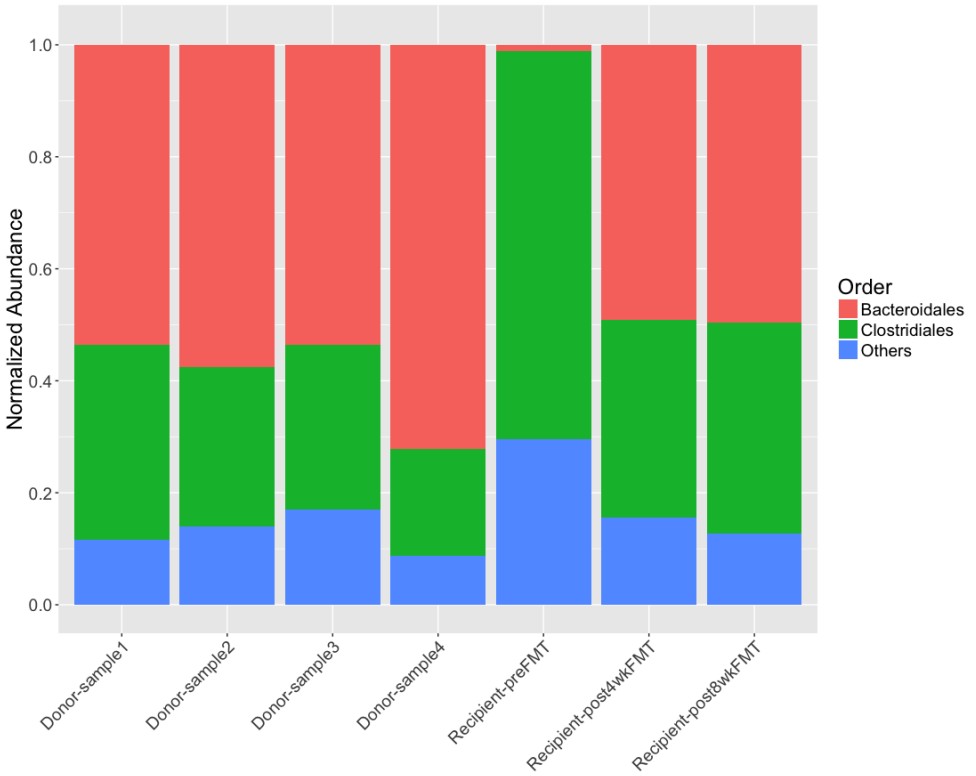

**Figure 5** Relative abundance of host bacteria at Order level in the FMT samples from FastViromeExplorer result using the 125,842 mVCs as reference, where abundance is normalized by the total abundance of viruses in the sample.

298,383 epipelagic and mesopelagic viral contigs collected from *Roux et al. (2016)*. Results show that FastViromeExplorer was able to successfully identify some contigs that resemble the original 483 scaffolds. For example, while using the 125,842 mVCs as reference, the most abundant mVC was "GOS2241_1000284", which according to blastn aligns with scaffold "NTLX01000031.1" with identity 76.33% and alignment length 562 bp. Another example was mVC with id "JGI25127J35165_1001802" which has similarity with the longest scaffold "NTLX01000001.1". In addition while using the GOV contig set (*Roux et al., 2016*) as reference, we identified an abundant contig with id "GOV_bin_1783_contig-100_1" with 98.35% identity and 2,004 bp alignment length with scaffold "NTLX01000307.1". These successful hits could subsequently be used to assemble the actual viral sequences found in the samples. Taken together, our results show that FastViromeExplorer can also be applied to detect and quantify viruses and phages in metagenomic samples taken from environmental samples, and the results are accurate if given a sufficiently complete reference database.

## CONCLUSION

In this paper, we develop a new tool (FastViromeExplorer) for detecting and quantifying viruses in metagenomic data. It is worth emphasizing that FastViromeExplorer can detect both viruses and phages depending on the reference database users deploy. As FastViromeExplorer can process millions of reads within minutes while having similar virus detection accuracy to the gold standard tool blastn, it empowers researchers that have limited computing power to process large metagenomic data within reasonable time. Similar to all other reference database tools, the limitation of FastViromeExplorer is that it cannot identify a virus or phage if a similar sequence is not present in the reference database; therefore, our tool cannot be used to identify or recover novel viruses that have no similarity to sequences in the reference database. Our preliminary results for the human microbiome and ocean environmental data highlight the pressing issue of building and/or extending the current viral sequence database for improving virus/phage detection and quantification in metagenomic data.

### Funding

The authors received no funding for this work.

### Competing Interests

The authors declare there are no competing interests.

### Author Contributions

- Saima Sultana Tithi conceived and designed the experiments, performed the experiments, analyzed the data, contributed reagents/materials/analysis tools, wrote the paper, prepared figures and/or tables, reviewed drafts of the paper.
- Frank O. Aylward analyzed the data, wrote the paper, reviewed drafts of the paper.
- Roderick V. Jensen and Liqing Zhang conceived and designed the experiments, analyzed the data, wrote the paper, reviewed drafts of the paper.

### Data Availability

FastViromeExplorer codebase: https://code.vt.edu/saima5/FastViromeExplorer.

Pre-computed Kallisto index files: http://bench.cs.vt.edu/FastViromeExplorer/.

FastViromeExplorer documentation: http://fastviromeexplorer.readthedocs.io/en/latest/.

### Supplemental Information

Supplemental information for this article can be found online at http://dx.doi.org/10.7717/peerj.4227#supplemental-information.

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
