# Peer review of "FastViromeExplorer: a pipeline for virus and phage identification and abundance profiling in metagenomics data"

_PeerJ, doi:10.7717/peerj.4227_

## Round 0.1 · original submission · Major Revisions

Both reviewers reported that this manuscript describes an important, useful tool to assist in virus discovery from metagenomic samples. But they also reported a number of shared concerns that need to be addressed before the manuscript is acceptable for publication.

The most significant concerns focus on the fact that FastViromeExplorer is a tool for detecting viruses with some degree of similarity to known viruses given that it is dependent on using a reference database of known virus sequences in order to determine that reads originated from a virus genome. As stated in the reviews, this fact needs to be stated much more explicitly in the manuscript, and the authors should also consider changing the name of the tool to better emphasize that it will not detect novel viruses with significant sequence divergence from known viruses. In addition, the reviewers were concerned that the results provided by FastViromeExplorer will be significantly dependent on the reference database employed, and suggest a number of ways that this dependence could be further explored to provide the reader with information on the sensitivity and limitations of the tool. While it is your decision as to how these issues are best addressed, they do represent a significant concern that need to be addressed in a revised manuscript.

The remaining concerns are minor, but should be carefully reviewed and addressed in a revised manuscript.

·

Basic reporting

I found the language in the manuscript to be sometimes a bit inaccurate and/or awkward, and I believe this should be improved so that the whole manuscript is entirely clear even for non-specialist readers (see examples below)

Experimental design

no comment

Validity of the findings

no comment

Additional comments

In this manuscript, the authors present a new tool (named FastViromeExplorer) destined to help detect and quantify known viruses in metagenome dataset. This tool is using a relatively recent approach (kallisto) to map metagenomic reads to a viral database, and then add a set of cutoffs to identify false positives (usually viral genomes which recruit reads only on a portion of the genome due to a repeat / shared region). The whole pipeline is provided as a stand-alone tool.
The main caveat of this tool is that it relies on databases so can’t provide any virus discovery. Given the state of the viral databases, this limitation should be made very clear as early as in the abstract (note that this can be presented as a trade-off with speed though, since any pipeline involving de novo assembly and identification of novel viral genomes would likely be much more computationally demanding and longer). As of now, the only mention of this bias is in the Conclusion section l. 354 “, the limitation of FastViromeExplorer is that it cannot identify a virus or phage if a similar sequence is not present in the reference database.”, which I consider is not enough for a non-specialist reader to understand how big/small of a problem this reliance on databases can be for different use case scenarios. The authors could also test / illustrate this database bias by generating simulated metagenomes including new viral genomes published since they created their database, and observe the maximum “distance” at which FastViromeExplorer is able to identify these viruses (i.e. how different from the database genomes a new virus genome can be), although I don’t see this additional benchmark as absolutely essential. More generally however, I would ask the authors to provide a much more detail description of which type of study would benefit from FastViromeExplorer (e.g. known pathogen detection) and which should use the more “standard” pipeline relying on de novo assembly (e.g. viral diversity exploration).
My other main concern is with the expected coverage computed as part of the mapping cutoffs, and calculated using a Poisson law. This doesn’t take into account biases linked to the PCR amplification of sequencing libraries, usually required for all/most “low input” cases, i.e. ng of input material, which is still relatively frequent for example in clinical setups (see e.g. PMID 23663384). In these PCR-amplified libraries, the coverage of a genome is not uniform (and less so when the number of PCR cycles increases), which means that the expected coverage cutoff used could wrongly exclude a number of “true” viral detections. Although I suspect there is no magic trick to solve this, I believe the authors should discuss these cases, and provide guidelines for how to use FastViromeExplorer when the user would not expect a uniform coverage.
Finally, I found the language in the manuscript to be sometimes a bit inaccurate and/or awkward, and I believe this should be improved so that the whole manuscript is entirely clear even for non-specialist readers (see examples below).

Additional comments:

l.23: “In a healthy human body there are estimated to be” feels a bit awkward to me, please rephrase.

l. 25-26: “diabetes […], and depression […] and cancer” Please remove the first “and” or reword the sentence.

l. 31: “composition and behavior of environmental microbiomes”. I am not sure a microbiome can have a “behavior”, maybe “metabolism” or “host interaction” could be more appropriate ?

l. 41: “annotating the taxonomy” should be “annotating genes for taxonomy” or “taxonomically affiliating”, but I don’t believe it is correct to say one “annotate” a “taxonomy”.

l. 56-58: “However, read assembly can be […] annotation.” Although I agree with the authors about the current challenges of assembling metagenomes, it would be great it they could add one or two references to back this claim.

l. 95: “30 million paired-end RNA-seq reads for the human transcriptome” shouldn’t it be “from a” human transcriptome “sample” rather than “for the human transcriptome” ?

l. 207: “through analyzing metagenomics data” should be “through the analysis of metagenomic data”.

l. 254: “is only one time process” should probably be modified to something like “is computed only once”.

l. 270: “As viruses are known to have fast mutation rates, Blastn and its variants (e.g., Blastp) are considered the “gold standard” approach to annotate viral sequences in metagenomic data but very time-consuming, having similar performance yet running much faster is highly desired for an annotation tool.” I disagree with this statement, and am not convinced it needs to be made at this point in the manuscript. First, the authors are comparing read mapping method, and correctly using blastn as an alternative to their tool, which has no real link to the high mutation rate observed in some viruses. On the other hand, blastp is used to annotate novel viruses (i.e. de novo assembly of viral genomes, usually only distantly related to viruses in the databases, and for which the only level at which sequence similarity can sometimes be observed is on the amino acid sequences). This is an entirely other approach, hence I believe referencing blastp in the context of this manuscript is misleading. Overall, I would suggest the authors remove this sentence entirely.

l. 285-287: The number of viruses identified seems very low, and I believe it would be helpful if the authors could add one sentence with the expected number of viruses in these samples, so that a reader can appreciate how well a database-relying approach using NCBI RefSeq worked in this case.

l. 310: Should provide information about the number of viral contigs identified with the larger DB (as it can be seen as a proxy for the number of viruses detected).

l. 311: “Therefore, the host information of the mVCs can be used to examine the annotation result.” I believe the authors should specify here how many sequences are associated with a host in this larger database and how these were derived. As it stands now, a reader could think that the two databases (RefSeq and IMG/VR) are equivalent in terms of host information, which would be misleading.

l. 317: “or or” should be “or”

l. 339: “Blastn” should be “blastn”

l. 350: “can annotate both” I disagree with the use of “annotate”, FastViromeExplorer does not provide any annotation, but instead can detect / quantify known viruses. This comment apply throughout the manuscript, hence variations of “annotate / annotation” should be modified elsewhere too when relevant (e.g. 352).

·

Basic reporting

I am happy with the writing style.

Experimental design

Good. I am happy with most of it other that the fact that this should have been tested with the full genbank viral database and not just the RefSeq as a database.

Validity of the findings

This method will be useful for people looking at known viruses in metagenomic datasets

Additional comments

The MS by Tithi et al describes a new rapid approach to identifying known viruses from metagenomics datasets. The approach relies on a reference viral database that is well curated e.g. GenBank or even the JGI viral metagenomics dataset and kallisto for rapid mapping of the reads. The output from the pipeline developed for FastViromeExplore provides a list of viruses that is sorted based on abundance. The latter is calculated based on number of reads.

I think this is a great tool for people who want to look for ‘known’ viruses in their dataset. However, I do see some major issues with the use of the word virome in the context of this software.
1) The tool described here is not a virome explore but a virus explore. The pipeline is heavily reliant on a nucleotide database. Over and over various studies in the last 10 years have shown that we have barely scratched the surface of viral nucleotide sequence space. Hence most of the virus discovery relives on protein sequence space.
2) The abstract “Identifying viruses and phages in metagenomic data has important implications in improving human health” is a bit odd. First of all phages are viruses and second identifying known viruses is not necessarily going to improve human health – it will trigger a response for clinicians and epidemiologists. Also with the approach you’ll only identify known viruses that may be emerging.
3) I am not sure refseq on it’s own is a good enough set of sequence for testing this tool. I would have tested in on the whole GenBank viral DB which includes variant. Roughly the RefSeq sequences represent a centroid in a 10% sequence variantion sequence space at a genome level. So it is likely that some reads may not be assigned to the correct “taxa” as it is missing from the database. Also you make a comment about in silico chimaeras during de novo assemblies. Read mapping to reference genomes also suppers from the same issue – that is assigned of reads to a wrong taxa. So you may want to address this.
4) Line 24-25: Please check this “Studies have shown that there are connections between human gut microbiome (viruses and bacteria) and diseases such as diabetes…” but as far as I am aware these studies have only looked at the role of bacteria not viruses.

I like this paper and think it has merit for publication in PeerJ. I also think that the authors may want to take a step back note that this tool is only for identifying viruses based on nucleotide sequences in the compiled database. I would change the name of the tool to FastVirusExplore a the word virome is totally misleading and this tool does not at all enable virome exploration.

---

## Round 0.2 · accepted · Accept

Thank-you for your resubmission.